# Changes in the Gulf Stream path over the last three decades

Antonio Sánchez-Román[1], Flora Gues[2], Romain Bourdalle-Badie[2], Marie-Isabelle Pujol[3], Ananda Pascual[1], Marie Drévillon[2]

[1]Mediterranean Institute for Advanced Studies, C/Miquel Marqués, 21, 07190 Esporles, Spain
[2]Mercator Ocean International, 2 Av. de l'Aérodrome de Montaudran, 31400 Toulouse, France
[3]Collecte Localisation Satellites, Parc Technologique du Canal, 8-10 rue Hermès, 31520 Ramonville-Saint-Agne, France

*Correspondence to*: Antonio Sánchez-Román (asanchez@imedea.uib-csic.es)

**Abstract.** The Gulf Stream transports warm waters from low to high latitudes in the North Atlantic Ocean, impacting Europe's climate. This study investigates the changing pattern of the Gulf Stream over the last three decades as observed in the altimetric record (1993–2022) using monthly-averaged altimetry maps together with the outputs from an ocean reanalysis product. The seasonal and yearly evolution of the coordinates (destabilization point) where the Gulf Stream starts to meander and to convert from a stable to an unstable detached jet is investigated. At seasonal scale, the location of this destabilization point presents zonal shifts displacing the Gulf Stream path to the north in summer and fall; and to the south in winter and spring. In addition, it presents variations at interannual scale and has varied by more than 1400 km in longitude showing meridional shifts of 300 km over the altimetric era: it exhibits a low-frequency remarkable shift westward and southward between 1995 and 2012. From that year, the destabilization point displacement inverses exhibiting a previously unreported migration eastward and northward that translates into a larger fraction of the stable detached jet in detriment of the unstable meandering jet. Changes in the Gulf Stream path impact both associated mesoscale Eddy Kinetic Energy and waters transported towards the subpolar North Atlantic. The observed shifts of the path destabilization point seem to be linked to North Atlantic Oscillation variability during winter that may play an important role: it presents a negative trend associated with a shift from a positive to a negative phase between 1995 and 2011; and an opposite behavior from a negative to a positive phase from that year until 2020 in agreement with the associated south-westward and north-eastward observed migration of the destabilization point.

## 1 Introduction

The Gulf Stream is part of the western boundary current system. It originates in the Gulf of Mexico and flows poleward close to the North American coast from the Straits of Florida to Cape Hatteras (Fig.1). Then, it leaves the continental margin and becomes a detached western boundary current flowing eastward as the Gulf Stream Extension (e.g., Joyce et al., 2009; Greatbatch et al., 2010). The Gulf Stream Extension carries near-surface warm waters from the subtropical to the subpolar North Atlantic (Guo et al., 2023) marking a transition from warm subtropical to cold subpolar waters (Joyce and Zhang,

2010; McCarthy et al., 2018) known as the Gulf Stream North Wall (GSNW). The GSNW is a sharp temperature front located to the north of the Gulf Stream that does not necessarily follows its path (Chi et al., 2019). The balance between these northward-flowing warm and shallow waters as part of the Gulf Stream and a southward cold and deep return

waterpath describes the Atlantic Meridional Overturning Circulation (AMOC, e.g., Buckley and Marshall, 2016; Lozier, 2019; Swingedouw et al., 2022). The AMOC accounts for nearly 90% of the total heat transport at 26.5ºN in the North Atlantic (Johns et al., 2011). Thus, it is a major driver of subpolar heat content changes (McCarthy et al., 2018). This makes the Gulf Stream play a paramount role in North Atlantic climate variability and change (Frankignoul et al., 2001, Joyce and Zhang, 2010; Srokosz et al., 2012; McCarthy et al., 2015; Lozier et al., 2019). Direct estimates of the GSNW are available

from 1955 (Joyce et al. 2000) and 1966 (Taylor and Stephens, 1980) onwards, allowing the analysis of the North Atlantic ocean circulation variability from decadal and multidecadal scales (McCarthy et al., 2018).

The Atlantic multidecadal variability is mainly due to internal ocean-driven variability associated with global and regional variations in precipitation and temperature, sea level fluctuations and hurricane activity (Delworth and Mann, 2000). However it could be also generated as a response to natural atmospheric variability (Clement et al., 2015) which is mainly

associated with the North Atlantic Oscillation (NAO). The NAO is the first mode of Atlantic atmospheric forcing and describes surface sea-level pressure differences between the Azores high and the subpolar low, and varies at quasi-decadal and multidecadal timescales (Da Costa & Colin de Verdiere, 2002; Gray et al., 2016; Årthun et al., 2017), impacting North Atlantic sea surface temperature patterns via air-sea heat exchanges (Hurrell et al., 2003; McCarthy et al., 2018; Osman et al., 2021).

The time-varying location of the Gulf Stream can be identified by using a constant sea surface height (SSH) contour from mapped absolute dynamic topography (ADT) from satellite altimetry to find snapshots of the current's path (Andres, 2016). The 25 cm SSH contour is commonly used (e.g. Lillibridge and Mariano, 2013; Rossby et al., 2014; Andres, 2016; Chi et al., 2021 and Guo et al., 2023). Other methods to identify the path of the Gulf Stream are based on the location of an isotherm at a given depth. Joyce et al. (2000; 2009) used the 15ºC isotherm at 200 m depth to define the region just to the north of strong

flow of the Gulf Stream that corresponds to the GSNW. This approach was followed by Frankignoul et al. (2001) and Seidov et al. (2019; 2021) to identify the latitude of Gulf Stream paths.

The variations in the Gulf Stream path exhibit two main modes: (i) wavelike fluctuations linked to the Gulf Stream meandering and instability,  and (ii) large-scale lateral shifts exhibiting seasonal and interannual changes (Frankignoul et al., 2001). Actually, western boundary currents are identified as eddy-rich regions where mean kinetic energy and available

potential energy from the mean flow are converted into mesoscale eddy kinetic energy (EKE) from baroclinic and barotropic instabilities. The low-frequency, interannual variability of the lateral shifts in the Gulf Stream position impacts on the global climate system as a whole (Guo et al., 2023) and can be linked to changes in climate-related oceanic phenomena such as El Niño-Southern Oscillation (Taylor et al., 1998), the AMOC (Joyce and Zhang, 2010) or the aforementioned atmospheric forcing (Wolfe et al., 2019) among others. Taylor et al. (1998) found that the Gulf Stream shifts were correlated with the

wintertime NAO during the time period spanning from 1966 to 1996 with high values of the NAO index (stronger

westerlies) favoring a northerly path 2–3 years later. Joyce et al. (2000) observed northward shifts of the Gulf Stream during positive phases of the NAO with lags of 1 year between 1954 and 1990. More recently, McCarthy et al. (2018) reported shifts in the Gulf Stream path coincident with NAO variations over both quasi-decadal and multidecadal timescales, having implications for linking the Gulf Stream path and AMOC.

The Gulf Stream path variability can be seen in gridded satellite altimetry and also in derived surface velocities as meridional shifts in the path of the Gulf Stream after Cape Hatteras (McCarthy et al., 2018). In this study, altimetry maps are used together with the outputs from an ocean reanalysis to assess the changing pattern of the Gulf Stream path over the last three decades, impacting both associated mesoscale EKE and waters transported towards the subpolar North Atlantic. To do that, the time-varying position of the path destabilization point where the Gulf Stream Extension converts from a stable,

detached jet to an unstable, meandering detached jet is investigated following the methodology described in Andres (2016). Furthermore, seasonal and interannual variability of the Gulf Stream path is assessed to investigate possible causes and consequences of observed Gulf Stream changes.

## 2 Methods

Daily maps of both ADT from satellite altimetry (product ref. no. 1, Table 1) and SSH from an ocean reanalysis product

(product ref. no. 2, Table 1) were averaged to produce monthly maps from January 1993 to December 2022. These maps have a spatial resolution of 1/4° and 1/12°, respectively. Then, the Gulf Stream path was identified with the 25 cm SSH contour according to e.g. Lillibridge and Mariano (2013); Rossby et al. (2014); Andres (2016), Chi et al. (2021) and Guo et al. (2023) from detrended ADT and SSH time series (Fig.1, panel a). The annual and semiannual cycles were kept in the time series to allow the analysis of the seasonal signal. Also, monthly-averaged geostrophic velocity fields derived from both

ADT and sea level anomaly (SLA) maps (product ref. no. 1, Table1) were used to estimate the surface velocity associated with the Gulf Stream paths. Geostrophic velocity anomalies derived from SLA maps were then used to compute the Gulf Stream surface EKE. EKE presents greater values in the vicinity of the main jets and currents such as the western boundary currents whereas it rapidly decreases elsewhere (von Schuckmann et al, 2016). Satellite gridded products miss part of the mesoscale variability due to coarser effective dynamical resolutions (Ballarotta et al., 2019). However, the interannual

variations in EKE can still be captured (Guo et al., 2022; 2023).

Variability of Gulf Stream paths was assessed on both a seasonal and yearly basis. Following Andres (2016), the 12 monthly mean paths for a given year were separated into 0.5° longitude bins and the variance of Gulf Stream position (latitude) in each bin was calculated. It can happen that the path in a given longitude bin describes a twisted route providing two or more latitudes. To overcome this, the most northerly latitude of the 25 cm SSH contour was used in the variance calculation

(Andres, 2016). This computation was also done for the Gulf Stream mean paths computed for 1993–2022 as a group (Fig. 1). The downstream distance (longitude) where the latitude's variance first reaches $0.42(°)^2$ (half of the maximum variance obtained for the aggregate) was defined as that year's path destabilization point. This is where the Gulf Stream converts from

a stable, detached jet to an unstable, meandering detached jet (Fig. 1, panel a). The confidence interval (at 95% confidence level) of the mean destabilization point was computed from the yearly destabilization point locations. A similar analysis was conducted for the seasonal assessment. Furthermore, the aforementioned computation was repeated from daily altimetry maps to compute the confidence interval (at 95% confidence level) of the yearly destabilization point location. To do that, the 30 daily paths for a given month were used to identify the month's path destabilization point. The 12 monthly destabilization points of a given year were then used to provide an estimation of the confidence interval for that year. Finally, a five-year running mean filter was applied to time series of the position of the destabilization point in the yearly assessment to avoid spurious signals due to changes in higher-frequency Gulf Stream variability.

**Table 1: data products used.**

| Product ref. no. | Product ID & type | Data access | Documentation |
|---|---|---|---|
| 1 | SEALEVEL_GLO_PHY_L4_MY_008_047;   Satellite observations | EU  Copernicus Marine Service Product (2023) | Quality information Document (QUID): Pujol et al. (2023) Product User Manual (PUM): Pujol (2022) |
| 2 | GLOBAL_MULTIYEAR_PHY_001_030; Ocean reanalysis | EU  Copernicus Marine Service Product (2022) | Quality information Document (QUID): Drévillon et al. (2022a) Product User Manual (PUM): Drévillon et al. (2022b) |

Also, the 12ºC isotherm (iso12) at 450 m depth was identified from the product ref. no. 2 (Table 1) and used to track the path of the Gulf Stream in the water column. This isotherm was chosen because it makes it possible to both limit short-term surface variations and follow the trajectory of the Gulf Stream more at depth than previous studies based on the temperature at 200m depth (e.g. Joyce et al., 2000; 2009).

## 3 Results

### 3.1 Transition of the Gulf Stream path to an unstable jet

The datasets and methods described above were used to characterize the mean and time-varying Gulf Stream path, and identify its transition to an unstable detached jet. Variance in Gulf Stream latitude increases abruptly around 65°W (inset in Fig. 1, panel a) and spreads around 1600 km out along the detached jet. In addition, a local minimum in variance is found to the west at around 70°W close to a node reported by e.g., Joyce et al. (2000).

The mean destabilization point of the monthly mean Gulf Stream paths (1993–2022) is located at coordinates close to 38°N and 66°W (Fig. 1, panel a). West of this location (i.e. near cape Hatteras), the path is stable exhibiting a relatively straight,

detached jet and thus low variance (inset in Fig. 1, panel a). Downstream from the destabilization point the path becomes unstable showing meanders that translate in high variance and associated mesoscale EKE.

The Gulf Stream is one of the regions with the strongest mesoscale energy in the global ocean (Chelton et al., 2011; Guo et al., 2023). It presents mean values larger than 2000 $cm^2/s^2$ downstream from 75°W where the Gulf Stream separates from the

continental margin and becomes the Gulf Stream Extension (Fig. 1, panel b). This area has an energetic mesoscale activity exhibiting strong eddy-mean flow interaction with significant along-stream variability (Kang and Curchitser, 2015, Guo et al., 2023).  The mean EKE (1993–2022) core, with values larger than 3000 $cm^2/s^2$, is observed in the surroundings of the Gulf Stream mean path. In addition, the zonally-maximum mean EKE exhibiting values larger than 4000 $cm^2/s^2$ is located close to the destabilization point where the Gulf Stream becomes unstable. These features are consistent with previous

observations both in the upstream and downstream parts of the flow (e.g., Kang and Curchitser, 2015).

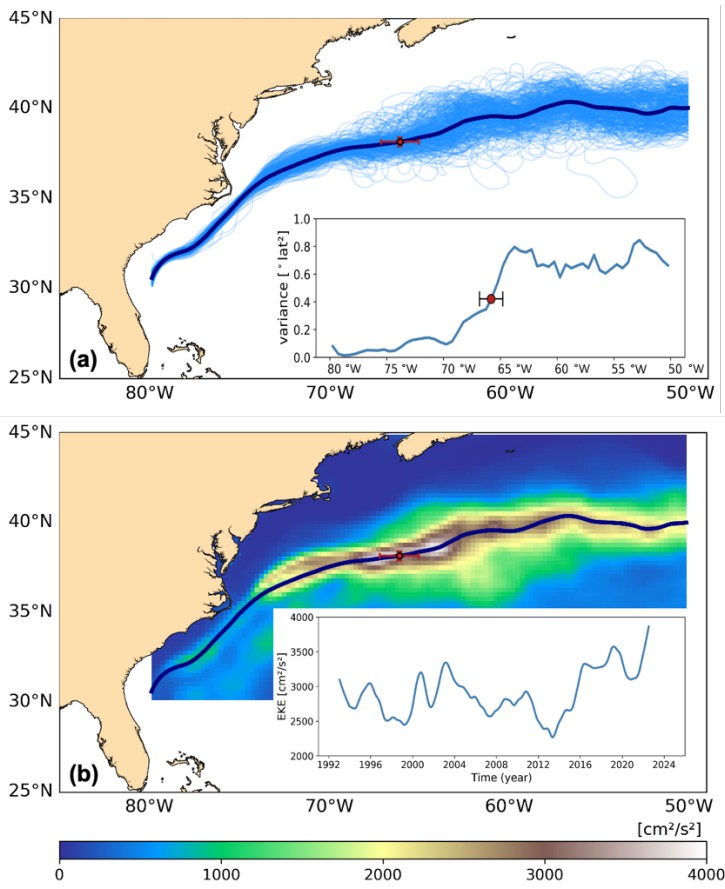

**Figure 1: panel a displays the Gulf Stream paths based on the 25 cm SSH contour from altimetry (product ref. no. 1) showing**
**monthly (pale-blue), and a 1993–2022 overall (blue) mean. The inset displays the variance in latitudinal position of the monthly mean Gulf Stream paths (1993–2022) as a function of downstream longitude. Panel b displays the 1993-2022 overall mean EKE in the Gulf Stream region and the 1993–2022 overall Gulf Stream mean path (blue). The inset shows the aggregated EKE associated**

with the Gulf Stream. Red dot in panels a & b indicates the mean destabilization point (see text for more details). The confidence interval (at 95% confidence level) of the destabilization point in both longitude and latitude is also displayed.

The aggregated (zonally and meridionally averaged) 1-year low-pass filtered surface geostrophic velocity associated with the Gulf Stream paths (figure not show) presents an overall negative linear trend over the period 1993–2012 with reduced speed exhibiting strong interannual variability at decadal and sub-decadal scale. This agrees with results reported by Dong et al. (2019) and Chi et al. (2021) from altimetry data for the same period. This fact translates into a recurring EKE decrease with values ranging from around 3000 $cm^2/s^2$ at the beginning of the altimetric era to close to 2200 $cm^2/s^2$ in 2012 (inset in Fig. 1,

panel b). From 2013 there is an inversion in the temporal evolution of the surface velocity linked to the Gulf Stream with an increasing speed until 2022 that promotes aggregated EKE values larger than 3500 $cm^2/s^2$ also showing interannual variability.

     Similar results were obtained from a computation using the climatological satellite product based on a steady number (two) of satellite missions (e.g., Sánchez-Román et al., 2023). Thus, this increasing EKE is not an artifact due to larger energy

promoted by a larger number of satellite missions used in the all-satellites product to generate the time series.

## 3.2 Interannual displacement of Gulf Stream

     Figure 2 shows the yearly evolution of the destabilization point in latitude (panel a) and longitude (panel b). Over the last three decades, the location of this destabilization point (red dots) has varied by more than 1400 km in longitude (i.e., between 57°W and 70°W) showing strong interannual variability (panel b). There has been an overall evolution of the

destabilization point of the Gulf Stream towards western longitudes particularly from 1995 to 2014, which agrees with the findings of Andres (2016) over the same period. However, from 2014 until 2022 an inversion in the temporal evolution of the destabilization point occurs showing a previously unreported displacement towards eastern longitudes. In addition, a meridional shift in the location of the destabilization point (panel a) of 300 km (i.e., between 37.7°N and 40.6°N) is observed promoting its displacement towards southern latitudes until 2014, and towards northern latitudes from that year until 2022.

These new findings expand the results reported in Andres (2016) and might have an impact on the physical properties of waters transported towards the subpolar eastern North Atlantic.

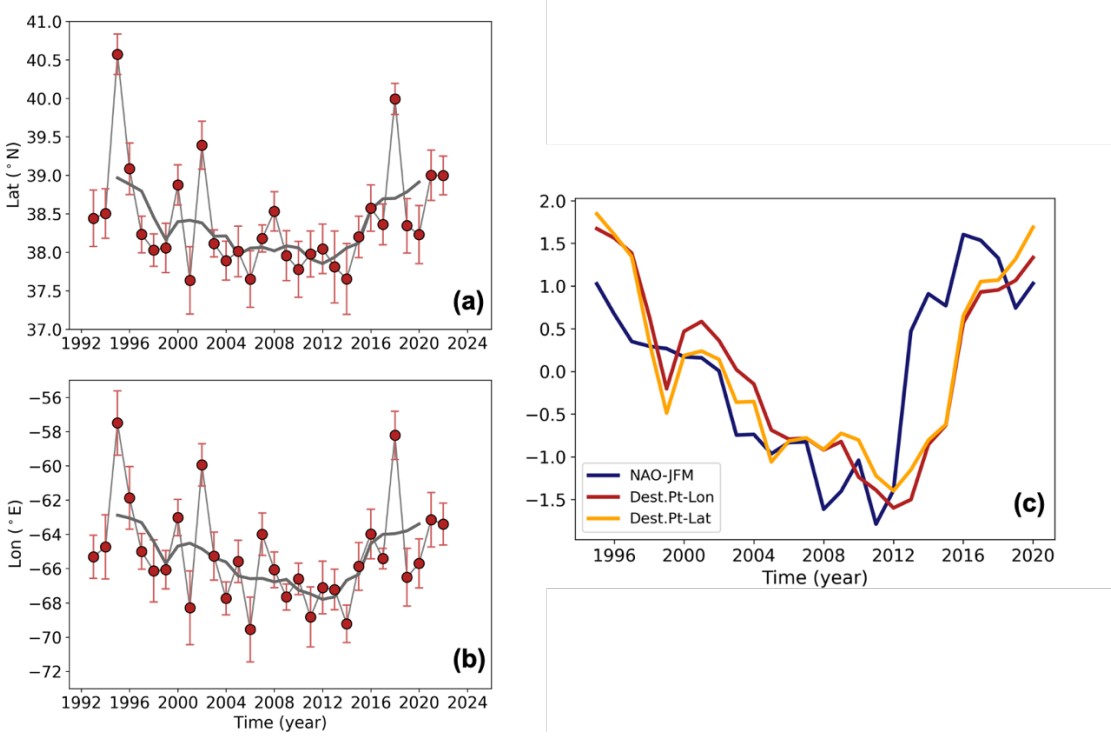

**Figure 2: yearly evolution of the destabilization point computed from altimetry data (product ref. no. 1) showing the latitude (panel a) and longitude (panel b) where the Gulf Stream becomes unstable. Solid grey line indicates the five-year running mean of the destabilization point. The confidence interval (at 95% confidence level) of the yearly position of the destabilization point in both longitude and latitude is also displayed. Panel c shows the standardized five-year running mean of the position (longitude - red line, latitude - orange line) of the destabilization point and the standardized five-year running mean of the seasonal mean NAO index during cold season (blue line).**

However, these results might be affected by spurious signals due to changes in higher-frequency Gulf Stream variability. To avoid this, the five-year running-mean of the position of the destabilization point was investigated (grey line in Fig. 2, panels a & b). The low-frequency variability of the Gulf Stream path indicates a westward and southward shift of the destabilization point from 1995 to 2012 that reverses towards northward and eastward shift from that year until 2020. This pattern agrees with the temporal evolution of the standardized five-year running mean of the annually averaged wintertime (January–March) NAO index (Fig. 2, panel c), that shows a Pearson linear correlation with the time-varying longitude (latitude) of the destabilization point of 0.70 (0.73) significant at 95%. In addition, the time-varying longitude of the low-frequency zonally-maximum EKE associated with the Gulf Stream path presents a linear correlation with the location of the destabilization point (figure not shown) of 0.88 exhibiting overall differences in longitude lower than 3 degrees. Thus, this temporal variability also matches the aforementioned time-varying surface velocities and derived mesoscale EKE associated with the Gulf Stream path giving support to the assessment of the low-frequency variability of the Gulf Stream.

## 3.3 Temperature signature of Gulf Stream pathway

Figure 3 panel a shows the mean Gulf Stream pathways estimated using the iso12 at 450 m depth for two representative two-year periods before (2008-2010) and after (2014-2016) the change in trend of the destabilization point, together with Gulf Stream trajectories estimated with the method based on SSH data for the same periods. The iso12 estimate of the Gulf Stream pathway is located north of the sea level estimate because the iso12 is a signature of the GSNW rather than of the center of the pathway (Chi et al., 2019; Seidov et al., 2021). The good correspondence between mean pathways estimated with the altimeter data and with the temperature data (panel a), indicates that the signal detected at the surface is also present in the subsurface. On both diagnostics a separation of the average Gulf Stream pathway between the two periods occurs near 66°W, which corresponds to the detected mean destabilization point. Downstream the mean pathways for the two periods converge. However, in the subsurface near 450 m, this convergence seems to occur upstream (near 62°W, purple box) on the Gulf Stream path than at the surface (east of 57°W, black box).

The meridional variability of monthly mean pathways estimated using the iso12 at 450 m depth for the two-year periods before and after the change of the destabilization point's trend (Fig. 3, panel b) also reflects the variation observed at the surface with the method based on SSH (Fig. 1, panel a). The spread in latitude of the monthly mean pathways estimated for the two periods decreases in the surroundings of the mean location of the destabilization point. This signature of a more stable pathway at this longitude thus confirms that the change in the destabilization point diagnosed from altimetry also has a signature in the subsurface on the temperature field.

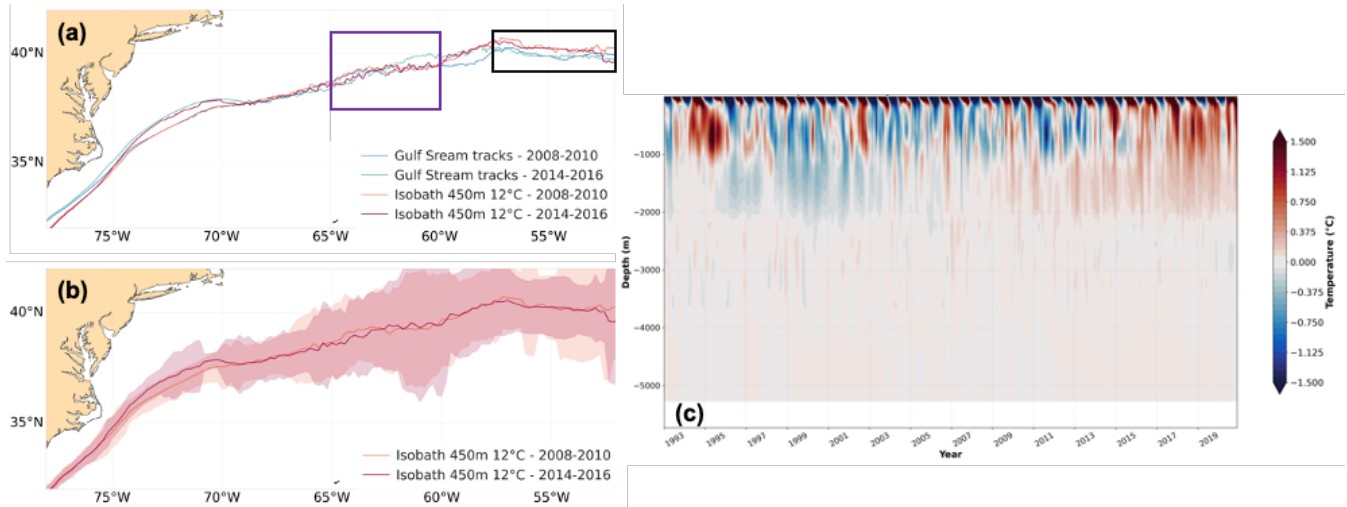

**Figure 3: panel a, the mean 12°C isotherm (iso12) for 2008-2010 (orange) and 2014-2016 (red) computed from ocean reanalysis data (product ref. no. 2) superimposed to the SSH derived Gulf Stream pathway computed from altimetry data (product ref. no. 1) for the same periods, blue and green respectively. Panel b, the mean and standard deviation (2 sigma) of monthly pathways estimated using the iso12 at 450 m depth computed from ocean reanalysis data (product ref. no. 2) for the periods 2008-2010 (orange) and 2014-2016 (red). Panel c displays the temporal evolution of temperature anomalies in the water column (°C) in the**

 surroundings (downstream) of the mean Gulf Stream's destabilization point (purple box) computed for the reference period 1993-2020.

Furthermore, the time evolution of the temperature in the upper part of the water column (Fig. 3, panel c) in the surroundings (downstream) of the Gulf Stream's destabilization point (purple box in panel a) exhibits a constant pattern over the first 1000 m and down to 2000 m. In addition, a strong negative hiatus was found in 2010, followed by a significant increase in temperature to become a positive anomaly in 2014. This increase may be due to both a long-term climate change and a change in the characteristics of the water masses. Thus, the analysis of the destabilization point of the Gulf Stream from SSH data could be a good indicator of the subsurface conditions (in the upper 1000 m of the water column) in the northeastern part south of the Grand Banks.

## 4 Discussion and conclusions

### 4.1 Seasonal and interannual variability of Gulf Stream paths

The Gulf Stream Extension displaces to the north in fall (exhibiting a relatively low baroclinic transport), and to the south in spring (Tracey and Watts, 1986) reaching its maximum baroclinic transport in early summer (Sato and Rossby, 1995). This seasonal pattern (Fig. 4, panels b,d, e; product ref. no. 1 in Table 1) is extended to summer (panels a & e) and wintertime (panels c & e), respectively. This is a novelty with respect to previous estimations (e.g. Lillibridge and Mariano, 2013) having an impact on the location of the destabilization point (inset in panel e): it shifts eastwards until 65°W in winter and 65.7°W in spring, the unstable meandering detached jet being shortened and located more to the south whereas it remains close to 66°W in summer and fall, the unstable jet being enlarged and located more to the north. The seasonal meridional shifts of the destabilization point are negligible with values ranging from 38.1°N in spring to 38.3°N in summer. On the contrary, this seasonal displacement of the path is not observed upstream of 70°W. Thus, the seasonal meridional shifts of the detached jet are accompanied by longitudinal seasonal variability of the destabilization point. This fact has an impact on the mesoscale EKE monitored in the Gulf Stream region that shows a clear seasonal variability (von Schuckmann et al., 2016) with maximum levels in the summer period (May to September) associated with a larger unstable meandering jet located more to the north and also more meso to sub-mesoscale activity (Ajayi et al., 2020); and minimum levels in winter (January) when the unstable jet is shorter and placed more to the south. These seasonal meridional fluctuations in Gulf Stream path position have important consequences for regional climate because the Gulf Stream transports considerable heat from the ocean at low latitudes to the atmosphere at high latitudes (Johns et al., 2011) and contributes to the distribution of biogeochemical properties in the North Atlantic Ocean (von Schuckmann et al., 2016).

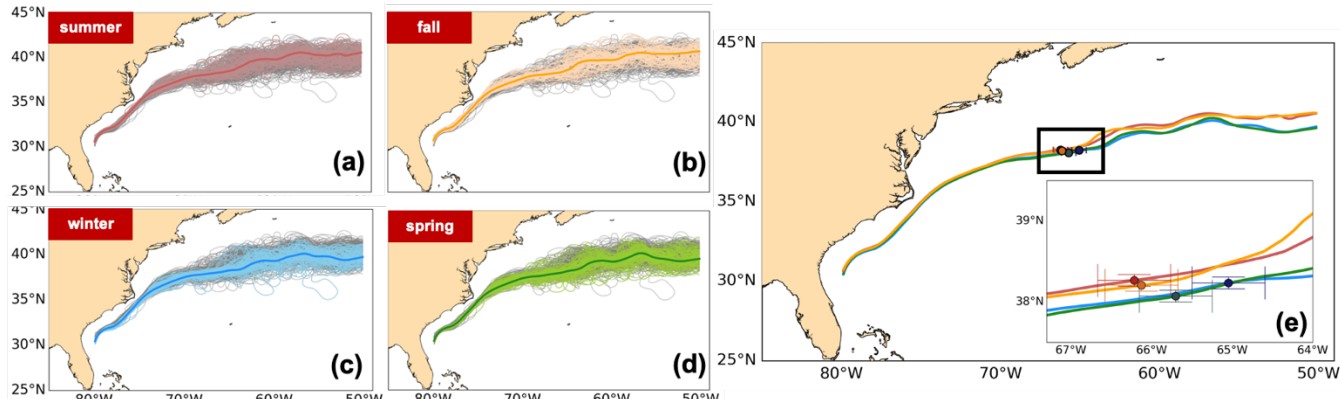

Figure 4: Gulf Stream paths based on the 25 cm SSH contour from altimetry (product ref. no. 1) showing monthly and a 1993–2022 overall seasonal mean for (a) summer (red, JAS), (b) fall (orange, OND), (c) winter (pale-blue, JFM) and (d) spring (green, AMJ). Panel (e) displays the 1993–2022 overall seasonal means with the mean location of the seasonal destabilization point. The inset displays a zoom of the region inside the black box. The confidence interval (at 95% confidence level) of the destabilization point in both longitude and latitude is also displayed.

In addition to the seasonal variability of Gulf Stream paths, the destabilization point of the detached jet exhibits a remarkable low-frequency shift westward between 1995 and 2012 accompanied by a southward shift of the jet. This promotes a shorter stable detached jet with time and thus eddying flows closer to the western boundary and the Middle Atlantic Bight (MAB) shelf that are widespread along a larger region of the North Atlantic. This proximity increases the probability of Gulf Stream-MAB interactions and have important consequences beyond a local increase in the EKE associated with the Gulf Stream (Andres, 2016). Warm core rings can spun off from the jet and bring salty and nutrient-reach deep waters to the euphotic zone at the shelfbreak front in the MAB leading to enhanced primary productivity (Zhang et al., 2013; Hoarfrost et al., 2019) and ecosystem changes (Gawarkiewicz et al., 2018). Monim (2017) reported an increase of 50% in the frequency of warm core rings formed annually in years 2000-2016 (overall, in agreement with the observed westward shift of the destabilization point) compared to 1977-1999 in the slope region south of New England having important effects on biogeochemical cycling (Hoarfrost et al., 2019).

In 2012 the destabilization point displacement reversal exhibits a previously unreported low-frequency migration eastward accompanied by a northward shift of the jet until 2020. This translates into a larger fraction of the stable detached jet in detriment of the unstable meandering jet that is likely to promote the depletion of the frequency of warm core ring intrusions onto the continental shelf and the probability of Gulf Stream-BAM interactions, in contrast with the increased interactions from the westward displacement observed in the recent past.

## 4.2 Impact of varying Gulf Stream stability on associated EKE

Guo et al. (2023) found a dominant component in mesoscale EKE associated with the Gulf Stream that co-varies with the meridional shift of the jet. Thus, migration of the destabilization point may have an impact on both the Gulf Stream's surface velocity and associated EKE. The low-frequency west-southward shift of the destabilization point observed between 1995

and 2012 is accompanied by a weakening of the jet (figure not shown) and associated mesoscale surface EKE (Fig. 1, panel b). Dong et al. (2019) attributed this velocity decrease to an increase in SSH to the north of the Gulf Stream mainly due to ocean warming.

The observed weakening of the jet over this period was explained by Renault et al. (2016) in terms of energy transfers from the ocean to the atmosphere over the Gulf Stream induced by the current feedback. It attenuates the wind surface stress inducing a positive surface stress curl opposite to the current vorticity that deflects energy from the Gulf Stream into the atmosphere and dampens eddies. It causes a mean pathway of energy from the ocean to the atmosphere (Renault et al., 2016a). Consequently, the current feedback promotes a slowdown of the jet and a drastic weakening of the EKE limiting the propagation of eddies. This mechanism could be fostered by the observed west-southward shift of the destabilization point.

On the other hand, the previously unreported low-frequency north-eastward shift observed from 2013 until 2020 promotes an increasing velocity with larger associated EKE (see Fig.1). Guo et al. (2023), based on Empirical Orthogonal Function (EOF) analysis, found a mode that suggests an enhancement in EKE when the Gulf Stream shifts to the North. Thus, the current feedback is likely to weaken in this period allowing energy transfers from the atmosphere to the ocean and the propagation of eddies. This would suggest a connection of the current feedback and net energy transfers between the atmosphere and the ocean with the observed meridional shifts of the jet and associated velocity rather than the variations in SSH linked to the ocean warming pointed out by Dong et al. (2019). However, the aforementioned increasing frequency of warm core ring intrusions onto the continental shelf observed during the low-frequency south-westward shift of the destabilization point can contribute to sea level rise through steric effect (Gawarkiewicz et al., 2018) reflecting a decreased sea level difference across the Gulf Stream (Sallenger et al., 2012) and a slowdown jet. The opposite is likely to account during the north-eastward displacement of the destabilization point when a larger fraction of the stable detached jet is observed in detriment of the unstable meandering jet. Thus, the Gulf Stream related processes could have an impact on sea level variability in the coastal region.

Furthermore, the global long-term change in surface mesoscale EKE found by Martinez-Moreno et al. (2021) might show that the Ocean EKE has experienced an increase. These changes in EKE also show that surface mesoscale diffusivities vary on climate time scales due to a coupling between large-scale climate variability and eddy mixing rates as a result of small amplitude changes in the large-scale flow (Busecke and Abernathey, 2019). These authors suggested that temporal variability in mesoscale mixing could be an important climate feedback mechanism due to the relevance of lateral mesoscale mixing for the ocean uptake of heat and carbon, and the distribution of oxygen and nutrients in the ocean, among others.

However, the underlying dynamics for the changes in the North Atlantic are not well understood and the mechanism behind correlations between EKE variability and Gulf Stream shifts are still unclear (Guo et al., 2022; 2023) and further investigations are needed.

## 4.3 External forcing of the Gulf Stream path destabilization

There are many factors, due to external forcing or reflecting internal variability, shaping the Gulf Stream system (Seidov et al., 2019) and thus, the observed shifts of the path destabilization point. The regimes of the Gulf Stream paths described above seem to be linked to NAO variability during winter (external forcing) that may play an important role. However, the relationships found between the Gulf Stream and the NAO depend on the analysis domain, the time period considered and the index used to define the Gulf Stream path position (Lillibridge and Mariano, 2013). Andres (2016) for instance found that the NAO index was uncorrelated at zero lag with the destabilization point of the detached Gulf Stream stating that the large- and regional-scale winds may not be directly responsible for the stability of the Gulf Stream jet. However, a maximum linear correlation of 0.80 (0.76) was found here between the NAO during winter and the time-varying longitude (latitude) of the destabilization point lagging by 1 yr (figure not shown). Thus, the Gulf Stream path seems to respond passively to the variability of the NAO during winter with a delay of a year at low frequencies. Frankignoul et al. (2001) stated that this delay is much shorter than expected from linear adjustment to wind stress changes and baroclinic Rossby wave propagation whereas it seems consistent with the assumption that the latitude of separation of the stable Gulf Stream is controlled by the potential vorticity of the recirculation gyres in the region. Actually, the wind stress curl (WSC) is responsible for the development and maintenance (via Ekman pumping) of the dipole of the two water gyres of the Gulf Stream system (Seidov et al., 2019) thus being coupled to the Gulf Stream dynamics (Renault et al., 2016). WSC together with the NAO stand out as the strongest external factors impacting the low-frequency Gulf Stream path variability at the sea surface on long-time scales.

The northward shift of the Gulf Stream path observed in the latest decade is likely to continue in the near future. It will probably impact on the zonal displacements of the destabilization point and may promote its migration to the east, and thus a larger fraction of the stable detached jet in detriment of the unstable meandering jet. Such changes in the position of the destabilization point seem to be being accompanied by a shift in the NAO index for winter. The observed time-varying Gulf Stream stability and associated ring dynamics may impact the frequency of warm core rings in the slope region south of New England and thus the upper ocean through changing events that drive the exchange of heat, nutrients and biogeochemical properties between the continental slope and outer shelf in the coming years.

### Data availability

Satellite observations and ocean reanalysis product are available from the Copernicus Marine Service web portal (https://doi.org/10.48670/moi-00148, last access: 5 December 2023, EU Copernicus Marine Service Product, 2023; https://doi.org/10.48670/moi-00021, last access: 15 December 2023, EU Copernicus Marine Service Product, 2022). The NAO index data is available from the US National Oceanic and Atmospheric Administration, Climate Prediction Center web portal (https://www.cpc.ncep.noaa.gov/products/precip/CWlink/pna/nao.shtml, last access: 6 December 2023).

## Author contribution

Conceptualization: ASR, MIP, AP, MD and RBB; satellite data processing: ASR and MIP; ocean reanalysis data processing: MD, RBB and FG; analysis and interpretation of results: ASR, MIP, RBB, AP and MD; paper writing: ASR, with input from all co-authors. All authors have read and agreed to the published version of the paper.

## Competing interests

The authors declare that they have no conflict of interest.

## Acknowledgements

This study has been conducted in the frame of the Copernicus Marine Service SL-TAC project. The Copernicus Marine Service, led by Mercator Ocean, is based on a distributed model of service production, relying on the expertise of a wide network of participating European organizations involved in operational oceanography. This work represents a contribution to the CSIC Interdisciplinary Thematic Platform (PTI) Teledetección (PTI-TELEDETECT), and it was carried out within the framework of the activities of the Spanish Government through the "Maria de Maeztu Centre of Excellence" accreditation to IMEDEA (CSIC-UIB) (CEX2021-001198).

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
