# Peer review of "Changes in the Gulf Stream path over the last three decades"

_State of the Planet, 2023_

## Author Comment (AC1)

**Reply to comments from the anonymous referee #1**

**General comments:**

The authors used altimetric data and an ocean reanalysis product to evaluate changes in the path and destabilization point of the Gulf Stream between 1993 and 2021. They demonstrated that the destabilization point has moved by 1400 km in longitude and 300 km in latitude over this period, in a process that is likely linked to the North Atlantic Oscillation variability.

Dear Madam/Sir,

we appreciate your comments which have been useful in improving the manuscript. Below we have responded to each of the specific comments and trust that these clarifications and amendments meet your approval. Please, notice that in the new version the time series have been extended to year 2022 because it is mandatory for OSR#8 publication so in this new version a time period of 30 years has been analyzed. Data for year 2022 was not included in the original version due to its unavailability in the Copernicus Catalogue when submitted.

Most of the results shown in the manuscript have already been shown by previously published papers, like the westward shift in the GS destabilization point (Andres, 2016), the relationship between NAO and GS path (Joyce et al, 2000), the changes in EKE (Chi et al, 2021) and seasonal meridional position (Tracey & Watts, 1986; Sato & Rossby, 1995). What seems novel is the recent eastward migration of the destabilization point following the westward migration described by Andres (2016), as well as the meridional shifts in the destabilization point and their seasonality. These points, however, are not highlighted as the main contributions of this manuscript (neither in the abstract nor during the results and discussion).

Following the reviewer suggestion, the new version of the manuscript has been focused on the eastern/meridional migration of the destabilization point and its seasonality, which have been highlighted in both the abstract and ant body of the text. In the following there are some examples of the sentences added to the new version:

Abstract

"The Gulf Stream transports warm waters from low to high latitudes in the North Atlantic Ocean, impacting Europe's climate. This study investigates the changing pattern of the Gulf Stream over the last three decades as observed in the altimetric record (1993–2022) using monthly-averaged altimetry maps together with the outputs from an ocean reanalysis product. The seasonal and yearly evolution of the coordinates (destabilization point) where the Gulf Stream starts to meander and to convert from a stable to an unstable detached jet is investigated. At seasonal scale, the location of this destabilization point presents longitudinal shifts displacing the Gulf Stream path to the north in fall and winter; and to the south in spring and summer. In addition, it presents variations at interannual scale and has varied by more than 1400 km in longitude

showing meridional shifts of 300 km over the altimetric era: it exhibits a low-frequency remarkable shift westward and southward between 1995 and 2012. From that year, the destabilization point displacement inverses exhibiting a previously unreported migration eastward and northward that translates into a larger fraction of the stable detached jet in detriment of the unstable meandering jet. Changes in the Gulf Stream path impact both associated mesoscale Eddy Kinetic Energy and waters transported towards the subpolar North Atlantic. The observed shifts of the path destabilization point seem to be linked to North Atlantic Oscillation variability during winter that may play an important role: it presents a negative trend associated with a shift from a positive to a negative phase between 1995 and 2011; and an opposite behavior from a negative to a positive phase from that year until 2020 in agreement with the associated south-westward and north-eastward observed migration of the destabilization point."

Results

"There has been an overall evolution of the destabilization point of the Gulf Stream towards western longitudes particularly from 1995 to 2014, which agrees with the findings of Andres (2016) over the same period. However, from 2014 until 2022 an inversion in the temporal evolution of the destabilization point occurs showing a previously unreported displacement towards eastern longitudes. In addition, a meridional shift in the location of the destabilization point (panel a) of 300 km (i.e., between 37.7°N and 40.6°N) is observed promoting its displacement towards southern latitudes until 2014, and towards northern latitudes from that year until 2022. These new findings expand the results reported in Andres (2016) and might have an impact on the physical properties of waters transported towards the subpolar eastern North Atlantic."

Discussion

"The seasonal meridional shifts of the destabilization point are negligible with values ranging from 38.1°N in spring to 38.3°N in summer. On the contrary, this seasonal displacement of the path is not observed upstream of 70°W. Thus, the seasonal meridional shifts of the detached jet are accompanied by longitudinal seasonal variability of the destabilization point. This fact has an impact on the mesoscale EKE monitored in the Gulf Stream region that shows a clear seasonal variability (von Schuckmann et al., 2016) with maximum levels in the summer period (May to September) associated with a larger unstable meandering jet located more to the north and also more meso to sub-mesoscale activity (Ajayi et al., 2020);  and minimum levels in winter (January) when the unstable jet is shorter and placed more to the south. These seasonal meridional fluctuations in Gulf Stream path position have important consequences for regional climate because the Gulf Stream transports considerable heat from the ocean at low latitudes to the atmosphere at high latitudes (Johns et al., 2011) and contributes to the distribution of biogeochemical properties in the North Atlantic Ocean (von Schuckmann et al., 2016)."

In addition, much of the discussion relies on points raised by Andres (2016) and the authors leave out of the discussion references that are extremely relevant to this topic, like Lilibridge & Mariano (2013) and Rossby et al (2014), which are only referenced for the sake of the definition of the GS path, or Dong et al (2019), which is only referenced to corroborate results. Other manuscripts like Renault et al (2016) and Seidov et al (2019) that could also contribute to the discussion are not referenced.

We thank the reviewer for the suggested papers, which have been used to improve the discussion in the new version. In the following there is an example of a paragraph from the discussion section in the new version:

"The observed weakening of the jet over this period was explained by Renault et al. (2016) in terms of energy transfers from the ocean to the atmosphere over the Gulf Stream induced by the current feedback. It attenuates the wind surface stress inducing a positive surface stress curl opposite to the current vorticity that deflects energy from the Gulf Stream into the atmosphere and dampens eddies. It causes a mean pathway of energy from the ocean to the atmosphere (Renault et al., 2016a). Consequently, the current feedback promotes a slowdown of the jet and a drastic weakening of the EKE limiting the propagation of eddies. This mechanism could be fostered by the observed west-southward shift of the destabilization point."

Renault, L., Molemaker, M. J., Gula, J., Masson, S., & McWilliams, J. C.: Control and stabilization of the gulf stream by oceanic current interaction with the atmosphere. Journal of Physical Oceanography, 46(11), 3439–3453. https://doi.org/10.1175/jpo-d-16-0115.1, 2016.
Renault, L., Molemaker, M. J. , McWilliams, J. C., Shchepetkin, A. F., Lemarié, F., Chelton, D., Illig, S.and Hall, A.: Modulation of wind work by oceanic current interaction with the atmosphere. J. Phys. Oceanogr., 46, 1685–1704, doi:10.1175/ JPO-D-15-0232.1, 2016a.

Overall, I think that the manuscript has a few interesting points, but most of its results and discussion rely heavily on previously published work. It could also benefit if underexplored points were highlighted in the results and discussion. For example, the conclusions shown in lines 247-251 seem relevant, but it was not clear to me how they were made, since the previous paragraph was confusing and lacked statistics and graphic visualizations. Although I do not think that this manuscript is a relevant contribution to the scientific understanding of the Gulf Stream understanding at this stage, I believe that it can make an impact if (1) the results that actually represent novel knowledge are highlighted (including visual and statistical representation) and (2) the discussion significantly improves by connecting the results to the existing knowledge rather than simply corroborating the results with previously published manuscripts.

Please notice that papers to be published in the OSR#8 are constrained to have a maximum of four figures. We consider that the figures included in the original version are representative of the main results addressed in the text. However, we agree with the reviewer that results must be accompanied by evidence so to solve this, we added a new panel to Figure 1 to show both the 1993-2022 overall mean eddy kinetic energy in the Gulf Stream region and its temporal evolution (aggregated values) to support the results discussed in the text. The same applies to Figure 3: we added a new panel with the temporal evolution of the temperature in the water column for the Gulf Stream region to support the results discussed in lines 247-249. We decided to remove the

sentences related to the EOF analysis (lines 239-246 and 249-251) since we cannot add more figures to the manuscript to support the reported outcomes.

In the new version, we highlighted the results representing novel knowledge related to the migration of the destabilization point at both seasonal and interannual frequencies, together with its implications in the transport of energy and nutrients in the North Atlantic Ocean. These results are based on extended time series with respect to previous studies. We also discussed them according to the actual knowledge of the Gulf Stream system.

**Specific comments:**

The most important method for GS path identification (25cm SSH contour) used in the manuscript is not referred to in the introduction. Why is that? Furthermore, the concept of GSNW, which is highly explored in the introduction, is not related to the GS path identification via altimetry. From the introduction, I would expect that the manuscript focused on the isothermal analysis of the GS path (i.e., analysis of GSNW), which is not the case. I would suggest modifying the introduction so that it more closely reflects the concepts and methods explored in the body of the manuscript.

The reviewer is right. We made a mistake in the previous version and wrongly identified the Gulf Stream path with the Gulf Stream North Wall. Thus, the introduction did not reflect the focus of the manuscript. This has been solved in the new version to avoid confusion. We have modified the introduction as follows to both include the different methods for Gulf Stream path identification according to your suggestion, and highlight that we are assessing the Gulf Stream path associated with the strong flow as identified via altimetry:

"The Gulf Stream is part of the western boundary current system. It originates in the Gulf of Mexico and flows poleward close to the North American coast from the Straits of Florida to Cape Hatteras (Fig.1). Then, it leaves the continental margin and becomes a detached western boundary current flowing eastward as the Gulf Stream Extension (e.g., Joyce et al., 2009; Greatbatch et al., 2010). The Gulf Stream Extension carries near-surface warm waters from the subtropical to the subpolar North Atlantic (Guo et al., 2023) marking a transition from warm subtropical to cold subpolar waters (Joyce and Zhang, 2010; McCarthy et al., 2018) known as the Gulf Stream North Wall (GSNW). The GSNW is a sharp temperature front located to the north of the Gulf Stream that does not necessarily follows its path (Chi et al., 2019)."

"The time-varying location of the Gulf Stream can be identified by using a constant sea surface height (SSH) contour from mapped absolute dynamic topography (ADT) from satellite altimetry to find snapshots of the current's path (Andres, 2016). The 25 cm SSH contour is commonly used (e.g. Lilibridge and Mariano, 2013; Rossby et al., 2014; Andres, 2016; Chi et al., 2021 and Guo et al., 2023). Other methods to identify the path of the Gulf Stream are based on the location of an isotherm at a given depth. Joyce et al. (2000; 2009) used the 15$^{\circ}$C isotherm at 200 m depth to define the region just to the

north of strong flow of the Gulf Stream that corresponds to the GSNW. This approach was followed by Frankignoul et al. (2001) and Seidov et al. (2019; 2021) to identify the latitude of Gulf Stream paths."

"The Gulf Stream path variability can be seen in gridded satellite altimetry and also in derived surface velocities as meridional shifts in the path of the Gulf Stream after Cape Hatteras (McCarthy et al., 2018). In this study, altimetry maps are used together with the outputs from an ocean reanalysis to assess the changing pattern of the Gulf Stream path over the last three decades, impacting both associated mesoscale EKE and waters transported towards the subpolar North Atlantic. To do that, the time-varying position of the path destabilization point where the Gulf Stream Extension converts from a stable, detached jet to an unstable, meandering detached jet is investigated following the methodology described in Andres (2016). Furthermore, seasonal and interannual variability of the Gulf Stream path is assessed to investigate possible causes and consequences of observed Gulf Stream changes."

Lines 156-158: The description of "matching time series" must be backed up by graphical/statistical representation, similar to what was presented for the relationship between the wintertime NAO index and destabilization point lat/lon.

We have added in the new version a sentence showing the linear correlation coefficient found between the zonally-maximum EKE associated with the Gulf Stream and the position of the destabilization point:

"In addition, the time-varying longitude of the low-frequency zonally-maximum EKE associated with the Gulf Stream path presents a linear correlation with the location of the destabilization point (figure not shown) of 0.88 exhibiting overall differences in longitude lower than 3 degrees. Thus, this temporal variability also matches the aforementioned time-varying surface velocities and derived mesoscale EKE associated with the Gulf Stream path giving support to the assessment of the low-frequency variability of the Gulf Stream."

Lines 162 and 171: The use of "before" and "after" is ambiguous. In line 162, it refers to time, so do you mean before and after the change in trends rather than the change in the destabilization point itself? In line 171, does it refer to time or space? If space, then it should be used "upstream and downstream" instead.

We aware that the concepts are ambiguous. In line 162 they refer to time related to the change in trends as mentioned by the reviewer. We have added this in the new version for clarity as follows:

"Figure 3 panel a shows the mean Gulf Stream pathways estimated using the iso12 at 450 m depth for two representative two-year periods before (2008-2010) and after (2014-2016) the change in trend of the destabilization point, together with Gulf Stream trajectories estimated with the method based on SSH data for the same periods."

They also refer to time related to the change in trends in line 171. To avoid confusion, we added this in the new version. Also, this paragraph has been reworded according to the next comment:

"The meridional variability of monthly mean pathways estimated using the iso12 at 450 m depth for the two-year periods before and after the change of the destabilization point's trend (Fig. 3, panel b) also reflects the variation observed at the surface with the method based on SSH (Fig. 1). The spread in latitude of the monthly mean pathways estimated for the two periods decreases in the surroundings of the location of the destabilization point. This signature of a more stable pathway at this longitude thus confirms that the change in the destabilization point diagnosed from altimetry also has a signature in the subsurface on the temperature field."

Lines 171-175: I did not understand what this paragraph describes, even compared with Fig 3b. What is the distribution of monthly mean pathways? Is it shown? What is the spatial extent of the standard deviation? How do you relate the signal in SSH to the signal in temperature? I suggest that this paragraph be rewritten for clarity.

We agree that the paragraph is confusing so it has been rewritten for clarity. Please, see response to the previous comment.

197-199: Are the meridional locations of the destabilization point really different between seasons? This is not clear from the map nor from statistics, which are not given.

We realize that they are not actually. We added the following sentence in the new version to highlight this:

"The seasonal meridional shifts of the destabilization point are negligible with values ranging from 38.1°N in spring to 38.3°N in summer. On the contrary, this seasonal displacement of the path is not observed upstream of 70°W. Thus, the seasonal meridional shifts of the detached jet are accompanied by longitudinal seasonal variability of the destabilization point."

Lines 210-219: There is plenty of literature reporting the influence of GS rings on the MAB, including those showing impacts on ecosystem changes and heatwaves, which could be explored in the discussion.

We have included in the new version of the manuscript discussion related to warm core rings from the Gulf Stream reaching the shelfbreak front in the MAB impacting the ecosystem in the region from Zhang et al., 2013, Hoarfrost et al., 2019, Gawarkiewicz et al., 2018 and Monim 2017. We associated their increase/decrease with the westward/eastward shift of the destabilization point. This paragraph has been modified in the new version as follows:

"In addition to the seasonal variability of Gulf Stream paths, the destabilization point of the detached jet exhibits a remarkable low-frequency shift westward between 1995 and 2012 accompanied by a southward shift of the jet. This promotes a shorter stable detached jet with time and thus eddying flows closer to the western boundary and the

Middle Atlantic Bight (MAB) shelf that are widespread along a larger region of the North Atlantic. This proximity increases the probability of Gulf Stream-MAB interactions and have important consequences beyond a local increase in the EKE associated with the Gulf Stream (Andres, 2016). Warm core rings can spun off from the jet and bring salty and nutrient-reach deep waters to the euphotic zone at the shelfbreak front in the MAB leading to enhanced primary productivity (Zhang et al., 2013; Hoarfrost et al., 2019) and ecosystem changes (Gawarkiewicz et al., 2018). Monim (2017) reported an increase of 50% in the frequency of warm core rings formed annually in years 2000-2016 (overall, in agreement with the observed westward shift of the destabilization point) compared to 1977-1999 in the slope region south of New England having important effects on biogeochemical cycling (Hoarfrost et al., 2019)."

"In 2012 the destabilization point displacement reversal exhibits a previously unreported low-frequency migration eastward accompanied by a northward shift of the jet until 2020. This translates into a larger fraction of the stable detached jet in detriment of the unstable meandering jet that is likely to promote the depletion of the frequency of warm core ring intrusions onto the continental shelf and the probability of Gulf Stream-BAM interactions, in contrast with the increased interactions from the westward displacement observed in the recent past."

Gawarkiewicz, G., Todd, R.E., Zhang, W., Partida, J., Gangopadhyay, A., Monim, M.-U.-H., Fratantoni, F., Malek Mercer, A. and Dent M.: The changing nature of shelf- break exchange revealed by the OOI Pioneer Array. Oceanography 31(1):60–70, https://doi.org/10.5670/oceanog.2018.110, 2018.

Hoarfrost A., Balmonte J.P., Ghobrial S., Ziervogel K., Bane J., Gawarkiewicz G. and Arnosti C.: Gulf Stream Ring Water Intrusion on the Mid-Atlantic Bight Continental Shelf Break Affects Microbially Driven Carbon Cycling. Front. Mar. Sci. 6:394. doi: 10.3389/fmars.2019.00394, 2019.

Monim, M.: Seasonal and Inter-Annual Variability of Gulf Stream Warm Core Rings from 2000 to 2016. Ph.D. thesis, University of Massachusetts, Dartmouth, 2017.

Zhang, W. G., McGillicuddy, D. J., and Gawarkiewicz, G. G.: Is biological productivity enhanced at the New England shelfbreak front? J. Geophys. Res. Oceans 118, 517–535. doi: 10.1002/jgrc.20068, 2013.

Lines 235-246: I do not see how this paragraph converses with the rest of the manuscript. What do you mean by "homogenous variations"? Why didn't the authors add any figures showing these results? From the following paragraph (lines 247-251) it seems like these results may be important to the overall understanding of the manuscript, but they are not shown.

This paragraph tries to connect the observed variability in the destabilization point at surface with changes in temperature of the upper part of the water column in the region. The term "homogenous variations" refers to the fact that changes in temperature in the surroundings of the destabilization point are observed in a region of the water column rather than at a specific depth. We are aware that the sentence is confusing so we reworded it in the new version as follows:

"Furthermore, the time evolution of the temperature in the upper part of the water column (Fig. 3, panel c) in the surroundings (downstream) of the Gulf Stream's

destabilization point (purple box in panel a) exhibits a constant pattern over the first 1000 m and down to 2000 m."

In addition, we added a new panel to figure 3 with the temporal evolution of the temperature in the water column for the Gulf Stream region to support the results discussed in the text.

On the other hand, the EOF analysis has been removed from the text in the new version because we cannot provide graphical support to the outcomes due to the limit of four figures for manuscripts to be published in the OSR#8 (see response to a previous comment)

**Technical corrections:**

Line 11: "Shift" instead of "displace".

The sentence has been removed in the new version.

Line 13: In the abstract, the authors mention the use of "outputs from a numerical model", but what they really use is a reanalysis product.

The reviewer is right. Thanks for point it out. The sentence has been reworded in the new version as follows:

"This study investigates the changing pattern of the Gulf Stream over the last three decades as observed in the altimetric record (1993–2022) using monthly-averaged altimetry maps together with the outputs from an ocean reanalysis product."

Line 28: "Gulf Stream play" instead of "Gulf Stream to play".

Done.

Line 57: "Northward shift" instead of "north shift".

Done.

Revise names in citations such as Rosby (should be Rossby) in line 77 and Kurchitser (should be Curshitser) in line 114.

Done. Thanks.

Line 78: What do you mean by "ADT (SSH)"? Following the pattern in the manuscript, acronyms in parentheses refer to the term immediately before, but this is not the case here.

We are aware that the sentence is confusing. We have reworded it in the new version as follows:

"Then, the Gulf Stream path was identified with the 25 cm SSH contour according to e.g. Lillibridge and Mariano (2013); Rossby et al. (2014); Andres (2016), Chi et al. (2021) and Guo et al. (2023) from detrended ADT and SSH time series (Fig.1, panel a)."

Line 86: "Following Andres (2016)" instead of "According to Andres (2016)".

Done.

Lines 100-101: Joyce did not use the 12°C isotherm at 450m to define the GS path in either of their cited papers.

Thanks for pointing it out. It was a misprint. We have updated the sentence as follows:

"Also, the 12°C isotherm (iso12) at 450 m depth was identified from the product ref. no. 2 (Table 1) and used to track the path of the Gulf Stream in the water column."

Line 112: "75°W" instead of "75°N"?

Done.

Line 113: "Gulf Stream Extension" instead of "GSNW"?

Done.

Line 153: 'Reverses" instead of "inverses".

Done.

Line 153: After applying a 5-yr running mean to a time series that runs until 2021, the final year is 2019, not 2021.

Thanks for the clarification. It has been updated in the text to avoid errors. Notice that in the new version the time series have been extended until 2022 so the 5-yr running mean's final year is 2020.

---

## Author Comment (AC2)

**Reply to comments from the anonymous referee #2**

This manuscript presents an analysis of Gulf Stream variability using gridded altimetry and an ocean reanalysis product. The primary focus is the location of the destabilization point first discussed by Andres (2016), and this manuscripts main result is an extension of Andres's calculation to later years. Andres (2016) had shown that the destabilization point was moving upstream, but Sánchez-Román et al. find that the destabilization point started moving back downstream around 2015. Further, the motion of the destabilization point is highly correlated with the NAO with a lag of 1-year. These are both fairly remarkable results, even if the manuscript's methodology is not particularly novel. A paper focused on investigating and explaining these results in greater detail would have been very interesting. Unfortunately, the present manuscript lacks focus and is not clearly written or organized. It also includes a great deal of extraneous and often unsupported material that makes it unsuitable for publication in its current form.

Dear Madam/Sir,

we appreciate your comments which have been useful in improving the manuscript. Below we have responded to each of the specific comments and trust that these clarifications and amendments meet your approval. The manuscript has been rewritten and reorganized for clarity, and now it has a more in depth discussion. In the new version, we highlighted the results representing novel knowledge related to the migration of the destabilization point at both seasonal and interannual frequencies, together with its implications in the transport of energy and nutrients in the North Atlantic Ocean. We also discussed them according to the actual knowledge of the Gulf Stream system. OSR papers are constrained to include a maximum of four figures so we added extra panels to the figures to further support the results. Please, notice that in the new version the time series have been extended to year 2022 because it is mandatory for OSR#8 publication so in this new version a time period of 30 years has been analysed. Data for year 2022 was not included in the original version due to its unavailability in the Copernicus Catalogue when submitted.

1. **Major Comments**

   There are long passages where results are quoted, but there are no figures or anything else to back them up. These results are practically meaningless without evidence. If the results are worth discussing, show a figure. If not, do not discuss them. The following passages should either be removed or be revised to include evidence:

   Lines 111–131

   Lines 235–251

Please notice that papers to be published in the OSR#8 are constrained to include a maximum of four figures. Thus, we cannot add new ones to the manuscript. We consider that the figures included in the original version are representative of the main results addressed in the text. However, we agree with the reviewer that results must be accompanied by evidence so to solve this, we added a new panel to Figure 1 to include both the 1993-2022 overall mean eddy kinetic energy in the Gulf Stream region and its temporal evolution (aggregated values) to support the results discussed in lines 111-131. The same applies to Figure 3: we added a new panel with the temporal evolution of the temperature in the water column for the Gulf Stream region to support the results discussed in lines 235-251. We decided to remove the sentences related to the EOF analysis since we cannot add more figures to the manuscript to support the results.

2. The usage of the term "Gulf Stream" is inconsistent, both internally and with oceanographic nomenclature. The current following the path shown in figure 1 is properly considered the Gulf Stream, but many things are called the "Gulf Stream" in the text that are not the Gulf Stream.

The reviewer is right. We made a mistake in the previous version and wrongly identified the Gulf Stream path with the Gulf Stream North Wall, and also with the North Atlantic Current. This has been solved in the new version for clarity. In the following, we response to each individual comment related to this.

1. Lines 33–34: It is stated that the Gulf Stream "becomes the North Atlantic Current". The North Atlantic Current and the Gulf Stream are geographically distinct. A fraction of the water that flows through the Gulf Stream eventually flows into the North Atlantic Current, but a large amount flows elsewhere as the North Atlantic Drift or as part of the Gulf Stream's recirculation gyres. Importantly, most of the water from the Gulf Stream ends up recirculating in the North Atlantic Subtropical Gyre and does not flow into the North Atlantic Current.

   This has been a misunderstanding. According to Stendardo et al. (2020): "The salinity import/freshwater export from/toward the subtropics is ensured by the North Atlantic Current (NAC) as a continuation of the Gulf Stream (e.g., Rossby, 1996) which supplies the subpolar gyre with warm and saline water from the subtropics as part of the upper branch of the AMOC." We understood that the Gulf Stream becomes the NAC but It is wrong. Thanks for the clarification. As the reviewer mentions, most of the water from the Gulf Stream recirculates in the North Atlantic Subtropical Gyre and not in the NAC so we removed the sentence in the new version to avoid confusion.

   Stendardo, I., Rhein, M., & Steinfeldt, R.: The North Atlantic Current and its volume and freshwater transports in the subpolar North Atlantic, time period 1993–2016. Journal of Geophysical Research: Oceans, 125, e2020JC016065. https:// 385 doi.org/10.1029/2020JC016065, 2020.

2. Line 10: "The Gulf Stream transports warm waters into the subpolar eastern North Atlantic … " The current that transports warm water into the subpolar North Atlantic is the aforementioned North Atlantic Current. The Gulf Stream does not have a direct subpolar connection.

Thanks for the clarification. The sentence in the abstract has been modified in the new version as follows to mention that the Gulf Stream carries warm waters from low to high latitudes in the North Atlantic:

"The Gulf Stream transports warm waters from low to high latitudes in the North Atlantic Ocean, impacting Europe's climate"

3. Lines 112–113: The Gulf Stream does not become the Gulf Stream North Wall (GSNW). The GSNW is part of the Gulf Stream, so the Gulf Stream cannot become the GSNW.

The reviewer is right. We mixed up the terms "Gulf Stream Extension" and "Gulf Stream North Wall". We have solved this misunderstanding in the new version and accordingly modify the sentence as follows:

"The Gulf Stream is one of the regions with the strongest mesoscale energy in the global ocean (Chelton et al., 2011; Guo et al., 2023). It presents mean values larger than 2000 cm$^2$/s$^2$ downstream from 75°W where the Gulf Stream separates from the continental margin and becomes the Gulf Stream Extension (Fig. 1, panel b)."

3. The usage of Gulf Stream North Wall (GSNW) also inconsistent and occasionally incorrect. The GSNW is the strong temperature front on the northern flank of the Gulf Stream. It is not "the path described by the Gulf Stream Extension" (lines 34–35) as it is north of the main core of the Gulf Stream. As noted in Chi et al. (2019), the GSNW and the main core of the Gulf Stream do not necessarily even follow the same path.

The reviewer is right. As we stated above, we made a mistake in the previous version and wrongly identified the Gulf Stream path with the Gulf Stream North Wall. This has been solved in the new version to avoid errors. The first paragraph of the introduction describing the Gulf Stream system has been reworded as follows:

"The Gulf Stream is part of the western boundary current system. It originates in the Gulf of Mexico and flows poleward close to the North American coast from the Straits of Florida to Cape Hatteras (Fig.1). Then, it leaves the continental margin and becomes a detached western boundary current flowing eastward as the Gulf Stream Extension (e.g., Joyce et al., 2009; Greatbatch et al., 2010). The Gulf Stream Extension carries near-surface warm waters from the subtropical to the subpolar North Atlantic (Guo et al., 2023) marking a transition from warm subtropical to cold subpolar waters (Joyce and Zhang, 2010; McCarthy et al., 2018) known as the Gulf Stream North Wall (GSNW). The GSNW is a sharp

temperature front located to the north of the Gulf Stream that does not necessarily follows its path (Chi et al., 2019)."

Lines 70–71: The Gulf Stream, not GSNW, converts from a stable to an unstable jet. The GSNW is a front, not a jet. Fronts are often associated with jets through thermal wind, but they are not the same thing.

See response to the previous comment. We have modified the sentence as follows to indicate that the Gulf Stream Extension converts from a stable to an unstable jet:

"To do that, the time-varying position of the path destabilization point where the Gulf Stream Extension converts from a stable, detached jet to an unstable, meandering detached jet is investigated following the methodology described in Andres (2016)."

4.  While gridded altimetry comes on a daily 1/4º grid, it does not have a temporal resolution of one day or a horizontal resolution of 1/4º. At the latitude of the Gulf Stream, the ground tracks used to construct the gridded produce are located approximately 2º apart and sampled roughly every 10 days. The values between the tracks and sample times are "filled in" using optimal interpolation. This produces smooth-looking fields, but can also invent spurious features and give the false impression of high precision. Ballarotta et al. (2019) estimates that gridded altimetry has an effective resolution of 150–200 km (e.g., 1.5º–2º) in the Gulf Stream region. As such, reporting locations obtained from gridded altimetry with sub-degree precision (as on line 107) is not meaningful. The authors reference Ballarotta et al. (2019) in noting that gridded altimetry misses some mesoscale features, but don't appear to acknowledge that the coarse resolution of altimetry may affect their estimates of the location of, for example, the destabilization point.

The effective spatial resolution showed by Ballarotta et al . (2019) refers to the DT-2018 version of the altimeter gridded products. Our study uses the up-to-date DT-2021 version which improves the previous one, including the effective spatial resolution, which is reduced to 100-150 km in the Gulf Stream region (Pujol et al., 2023). As discussed in Ballarotta et al . (2019) the effective spatial resolution is computed for maps constructed with three altimeters (CryoSat-2, HY-2, Jason-2) over the period 12 April 2014–31 December 2015 being Saral/AltiKa data used as an independent dataset. These authors state that "we believe that this assessment of the spatial resolution based on maps constructed with three altimeter missions may be considered a reasonable averaged estimate since about three altimeter missions are used in the merging for the CMEMS products 70 % of the time over the period 1 January 1993–15 May 2017." Nevertheless, the multi-mission gridded product is computed with a satellite constellation including all the available altimeters at a given time (ranging from two to seven over the period considered in this study; see, e.g., Fig. 1 in International Altimetry Team, 2021; Morrow et al., 2023). As a consequence, the errors are not constant in time since they depend on the

number of satellites used and we can reasonably consider that this error is reduced when more than 3 altimeters are used (i.e. over the last decade).

We also want to highlight that the manuscript mainly focuses on the seasonal/interannual variability and long-term longitudinal/latitudinal migration of the destabilization point rather than on its location at a given time. The text also include different elements to aware the users of the limitation of the methodology and quantify the errors of estimation of the location of the destabilization point : together with the location for the 1993-2022 mean and also for the yearly assessment, we provide its confidence interval at 95% confidence level computed from monthly-averaged and daily data, respectively (see text in the new version for details and response to the next comment). This confidence interval has a value of 1 degree in longitude and 0.24 degrees in latitude for the 1993-2022 mean path destabilization point; and ranges respectively between 0.6-2.16 degrees and between 0.17-0.47 degrees for the yearly location of the destabilization point. As we mention in lines 83-85 in the original version, "Satellite gridded products miss part of the mesoscale variability due to coarser effective dynamical resolutions (Ballarotta et al., 2019). However, the interannual variations in EKE can still be captured (Guo et al., 2022; 2023)."

We are aware that we cannot provide such a precise location of the mean destabilization point so we modified the sentence of former line 107 as follows:

"The mean destabilization point of the monthly mean Gulf Stream paths (1993–2022) is located at coordinates close to 38°N and 66°W (Fig. 1, panel a)."

International Altimetry Team: Altimetry for the future: Build- ing on 25 years of progress, Adv. Space Res., 68, 319–363, https://doi.org/10.1016/j.asr.2021.01.022, 2021.

Morrow, R., Fu, L. L., Rio, M. H., Ray, R., Prandi, P., Le Traon, P.-Y., Benveniste, J.: Ocean Circulation from Space, Surv, Geo- phys., https://doi.org/10.1007/s10712-023-09778-9, 2023.

Pujol, M-I, Taburet G., and SL-TAC team: EU Copernicus Marine Service Product Quality Information Document for the 370 Global Ocean Gridded L4 Sea Surface Heights And Derived Variables Reprocessed 1993 Ongoing, SEALEVEL_GLO_PHY_L4_MY_008_047, Issue 8.2, Mercator Ocean International, https://catalogue.marine.copernicus.eu/documents/QUID/CMEMS-SL-QUID-008-032-068.pdf. Last access: 4 April 2023, 2023.

Sánchez-Román, A., Pujol, M. I., Faugère, Y., and Pascual, A.: DUACS DT2021 reprocessed altimetry improves sea level retrieval in the coastal band of the European seas, Ocean Sci., 19, 793–809, https://doi.org/10.5194/os-19-793-2023, 2023.

5.  Lines 94–95: Without information about how the confidence interval computed there's no way for the reader to determine if they trust it.

The confidence interval of the yearly location of the destabilization point was computed from daily data as follows: we identified the daily path of the Gulf Stream from the 25 cm SSH contour from daily altimetry maps for the whole time period considered (30 years). Then, the 30 daily paths for a given month were separated into 0.5° longitude bins and the variance of Gulf Stream position (latitude) in each bin was calculated. The downstream distance (longitude) where the latitude's variance first reaches $0.42(°)^2$ was defined as that month's path destabilization point. The location (longitude and latitude) of the 12 monthly destabilization points of a given year were used to provide an estimation of the confidence interval (at 95% confidence level) of the destabilization point for that year. The same was applied to compute the confidence interval of the 1993-2022 mean destabilization point. This has been summarized in the new version as follows:

"The confidence interval (at 95% confidence level) of the mean destabilization point was computed from the yearly destabilization point locations. A similar analysis was conducted for the seasonal assessment. Furthermore, the aforementioned computation was repeated from daily altimetry maps to compute the confidence interval (at 95% confidence level) of the yearly destabilization point location. To do that, the 30 daily paths for a given month were used to identify the month's path destabilization point. The 12 monthly destabilization points of a given year were then used to provide an estimation of the confidence interval for that year."

6.  Figure 2b: The value given for 1994 in figure 2b is approximately 5º to the west of that in Andres (2016) figure 3. What is the source of this disagreement?

Andres (2016) identifies the downstream distance (longitude) where the latitude's variance first reaches $0.5(°)^2$ as the path destabilization point for a given year. Here, we use indeed the half of the maximum variance obtained for the aggregate (1993-2022), that is $0.42(°)^2$. As a consequence, we obtain a different yearly location of the destabilization point than that reported by Andres (2016). However, to check our method, we repeated the analysis using the value of $0.5(°)^2$ and we obtained quite similar locations to those reported by Andres (2016). In the following there is the paragraph explaining the method. We removed the reference of Andres (2016) to avoid confusion:

"Following Andres (2016), the 12 monthly mean paths for a given year were separated into 0.5° longitude bins and the variance of Gulf Stream position (latitude) in each bin was calculated. It can happen that the path in a given longitude bin describes a twisted route providing two or more latitudes. To overcome this, the most northerly latitude of the 25 cm SSH contour was used in the variance calculation (Andres, 2016). This computation was also done for the Gulf Stream mean paths computed for 1993–2022 as a group (Fig. 1). The downstream distance (longitude) where the latitude's variance first reaches

0.42(°)$^2$ (half of the maximum variance obtained for the aggregate) was defined as that year's path destabilization point. This is where the Gulf Stream converts from a stable, detached jet to an unstable, meandering detached jet (Fig. 1, panel a)."

7. Section 4: This section is labeled "Discussion and conclusions", but it is almost entirely more results rather than a discussion or a conclusion.

This point has been also raised by the reviewer #1. The section has been accordingly updated to highlight the implications of the recent eastward migration of the destabilization point following the westward migration described by Andres (2016), as well as the meridional shifts in the destabilization point together with their seasonality. In the following there are some examples of paragraphs added to this section including discussion, following also reviewer#1 comments:

Section 4.1: Seasonal and interannual variability of Gulf Stream paths

[revised manuscript text omitted]

8. Section 4.1: Figure 2b shows that the destabilization points migrates by more than 10º on interannual timescales. Against this background, the ~1º seasonal shifts are not meaningful. Indeed, the confidence intervals in figure 4e are barely non-overlapping.

The main idea of this section is to highlight the meridional seasonal fluctuations of the Gulf Stream Extension to the north in summer/fall, and to the south in winter/spring rather than the location of the seasonal destabilization point. The fact that this seasonality is only observed in the Gulf Stream extension (detached jet east of 70°W) makes a reduced longitudinal migration of the destabilization point at seasonal scales when compared with the long-term one. However, it is not negligible so we strongly think that it should be kept in the text. On the contrary, the meridional displacements of the seasonal destabilization point are negligible with respect to the longitudinal variability so we added the following sentence to the new version:

"The seasonal meridional shifts of the destabilization point are negligible with values ranging from 38.2°N in spring to 38.3°N in summer. On the contrary, this meridional displacement of the 1993–2022 mean path is not observed upstream of 70°W. This makes the observed seasonal shifts of the jet to promote longitudinal seasonal variability of the destabilization point."

9. Lines 259–263: The fact that Andres (2016) found a 5 year lag between the destabilization point does not justify a 5-year running mean filter. A centered running mean has no effect on the phase of a time series, so a 5-year lag in the original time series would remain a 5-year lag in the smoothed time series.

The reviewer is right. The sentence in its present form leads to confusion. We tried to explain that the uncorrelation at zero lag between NAO and the destabilization point found by Andres (2016) supports the approach followed here of assessing low-frequency variability of Gulf Stream path, which is based on a five-year running mean filter. Actually, this five-year running mean filter comes from the filter applied to the NAO by the NOAA /National Weather Service, available at:

https://www.cpc.ncep.noaa.gov/products/precip/CWlink/pna/JFM_season_nao_index.shtml

The sentence has been removed in the new version to avoid confusion about the filter applied.

**Minor Comments**

1. Lines 26–27: The authors should clarify that the AMOC accounts for 90% of the heat transport at 26.5ºN (the latitude of the RAPID array). There is no reason to expect that this holds true at other latitudes.

   Thanks for the clarification. It has been included in the new version as follows:

   "The AMOC accounts for nearly 90% of the total heat transport at 26.5ºN in the North Atlantic (Johns et al., 2011)."

2. Lines 27–28: It is not clear what "subpolar planetary heat exchange" means. Is it heat exchange on a planetary scale?

   We tried to emphasize in this sentence that the AMOC is a major driver of subpolar heat changes on a planetary scale. We have reworded the sentence in the new version for clarity as follows:

   "The AMOC accounts for nearly 90% of the total heat transport at 26.5ºN in the North Atlantic (Johns et al., 2011). Thus, it is a major driver of subpolar heat content changes (McCarthy et al., 2018)."

3. Line 49: Not clear what "instability" means in this context. Why are lateral shifts more associated with instability than meandering?

   There is a misprint in the sentence. We wanted to indicate that the variations in the Gulf Stream path are due to wavelike fluctuations linked to the Gulf Stream meandering and instability; and also to large-scale lateral shifts of the path. We have modified the sentence for clarity:

   "The variations in the Gulf Stream path exhibit two main modes: (i) wavelike fluctuations linked to the Gulf Stream meandering and instability, and (ii) large-scale lateral shifts exhibiting seasonal and interannual changes."

4. Line 62–63: Joyce used the 15ºC isotherm at 200 m depth. As written, the text suggests that the 15ºC isotherm is equivalent to the 200-m temperature.

   The sentence has been reworded and added to a new paragraph describing the different methodologies to identify the location of the Gulf Stream path:

   "The time-varying location of the Gulf Stream can be identified by using a constant sea surface height (SSH) contour from mapped absolute dynamic topography (ADT) from satellite altimetry to find snapshots of the current's path (Andres, 2016). The 25 cm SSH contour is commonly used (e.g. Lillibridge and Mariano, 2013; Rossby et al., 2014; Andres, 2016; Chi et al., 2021 and Guo et al., 2023). Other methods to identify the path of the Gulf Stream are based on the location of an isotherm at a given depth. Joyce et al. (2000; 2009) used the 15ºC isotherm at 200 m depth to define the region just to the north of strong flow of

the Gulf Stream that corresponds to the GSNW. This approach was followed by Frankignoul et al. (2001) and Seidov et al. (2019; 2021) to identify the latitude of Gulf Stream paths."

5. Line 63: Joyce's index locates the GSNW which, as noted previously, is not necessarily colocated with the region of strong flow.

The reviewer is right (see response to the previous comment). We updated the sentence as follows for clarity:

"Joyce et al. (2000; 2009) used the 15°C isotherm at 200 m depth to define the region just to the north of strong flow of the Gulf Stream that corresponds to the GSNW."

6. Figure 3: It would help to indicate the regions discussed in the text on the figure.

The Figure 3 has been updated and now includes the location of the regions discussed in the text.

7. Line 173: The range over which the standard deviation decreases between the two periods is tiny—only 1º, which is below the effective resolution of gridded altimetry. What is the reader supposed to take away from this result?

The analysis of the temperature signature of the Gulf Stream path is conducted through the assessment of the reanalysis product, which has a spatial resolution of 1/12 degrees, that is enough to investigate differences in a spatial range of 1 degree.

8. Lines 210–211: Should "low-frequency remarkable shift" be "remarkable low-frequency shift"?

Yes. It has been reworded in the new version as follows:

"In addition to the seasonal variability of Gulf Stream paths, the destabilization point of the detached jet exhibits a remarkable low-frequency shift westward between 1995 and 2012 accompanied by a southward shift of the jet."

9. Lines 211–203: This is a very confusing sentence. What is widespread along a larger fraction of the North Atlantic? Larger fraction than what?

We are sorry for the confusing wording of the sentence. We tried to highlight that a westward shift of the destabilization point promotes a shorter stable jet and, therefore, eddying flows closer to the western boundary that are widespread along a larger region of the North Atlantic. We reworded the sentence as follows:

"This promotes a shorter stable detached jet with time and thus eddying flows closer to the western boundary and the Middle Atlantic Bight (MAB) shelf that are widespread along a larger region of the North Atlantic."

10. Lines 224–225: How do we see from the results presented here that the shift of the destabilization point was accompanied by a weakening of the jet?

As it was aforementioned, we are constrained to include a maximum of four figures in the manuscript. Thus, we decided to not include the aggregated velocity associated to the jet. However, the figure 1 in the new version displays the time series of the aggregated EKE associated with the jet which is indicative of its velocity. The sentence has been reworded as follows:

 "The low-frequency west-southward shift of the destabilization point observed between 1995 and 2012 is accompanied by a weakening of the jet (figure not shown) and associated mesoscale surface EKE (Fig. 1, panel b)."

11. Lines 230–231: How do changes in EKE show that surface mesoscale diffusivities are largely influenced by climate variability?

Busecke and Abernathey (2019) found strong evidence that mixing rates in the ocean vary on interannual and longer time scales in many regions of the global ocean. They stated that the observed mixing rates suggest a coupling between large-scale climate variability and eddy mixing rates due to small amplitude changes in the large-scale flow. They suggested that temporal variability in mesoscale mixing could be an important climate feedback mechanism due to the importance of lateral mesoscale mixing for the ocean uptake of heat and carbon, the distribution of oxygen and nutrients in the ocean, ENSO dynamics, and water mass formation.

We have updated the sentence to clarify this issue as follows:

"These changes in EKE also show that surface mesoscale diffusivities vary on climate time scales due to a coupling between large-scale climate variability and eddy mixing rates as a result of small amplitude changes in the large-scale flow (Busecke and Abernathey, 2019). These authors suggested that temporal variability in mesoscale mixing could be an important climate feedback mechanism due to the relevance of lateral mesoscale mixing for the ocean uptake of heat and carbon, and the distribution of oxygen and nutrients in the ocean, among others."

Busecke, J. J., & Abernathey, R. P.: Ocean mesoscale mixing linked to climate variability. Science Advances, 5(1), eaav5014. https://doi. org/10.1126/sciadv.aav5014, 2019.

12. The final paragraph of the manuscript (Lines 270–279) does not seem to follow from the results presented in the paper. Indeed, it mostly summarizes background material and would fit better in the introduction. It would be worth, however, ending the paper with a proper conclusion.

We have updated the sentence as follows:

"The northward shift of the Gulf Stream path observed in the latest decade is likely to continue in the near future. It will probably impact on the zonal displacements of the destabilization point and may promote its migration to the east, and thus a larger fraction of the stable detached jet in detriment of the unstable meandering jet. Such changes in the position of the destabilization point seem to be being accompanied by a shift in the NAO index for winter. The observed time-varying Gulf Stream stability and associated ring dynamics may impact the frequency of warm core rings in the slope region south of New England and thus the upper ocean through changing events that drive the exchange of heat, nutrients and biogeochemical properties between the continental slope and outer shelf in the coming years. "

**Technical Corrections**

1. The word "isobath" is used to mean depth, but it does not. An isobath is a contour of constant distance between the surface and the bottom. It is equivalent to a topographic contour. Replace with "depth" on lines 62 and 100.

   done

2. Replace "inverses" with "reverses" on line 153.

   done

3. Replace "inverses" with "reversal" on line 216.

   done